# How Paretic and Non-Paretic Ankle Muscles Contract during Walking in Stroke Survivors: New Insight Using Novel Wearable Ultrasound Imaging and Sensing Technology

**DOI:** 10.3390/bios12050349

**Published:** 2022-05-18

**Authors:** Pei-Zhao Lyu, Ringo Tang-Long Zhu, Yan To Ling, Li-Ke Wang, Yong-Ping Zheng, Christina Zong-Hao Ma

**Affiliations:** 1Department of Biomedical Engineering, The Hong Kong Polytechnic University, Hong Kong SAR 999077, China; pei-zhao.lyu@connect.polyu.hk (P.-Z.L.); ringo-tanglong.zhu@connect.polyu.hk (R.T.-L.Z.); jane.yt.ling@connect.polyu.hk (Y.T.L.); akewanglike@hotmail.com (L.-K.W.); yongping.zheng@polyu.edu.hk (Y.-P.Z.); 2Research Institute for Smart Ageing, The Hong Kong Polytechnic University, Hong Kong SAR 999077, China

**Keywords:** stroke, gait, tibialis anterior (TA), medial gastrocnemius (MG), dynamic ultrasound image, muscle activity, electromyography (EMG), mechanomyography (MMG)

## Abstract

Abnormal muscle tone and muscle weakness are related to gait asymmetry in stroke survivors. However, the internal muscle morphological changes that occur during walking remain unclear. To address this issue, this study investigated the muscle activity of the tibialis anterior (TA) and medial gastrocnemius (MG) of both the paretic and non-paretic sides during walking in nine stroke survivors, by simultaneously capturing electromyography (EMG), mechanomyography (MMG), and ultrasound images, and using a validated novel wearable ultrasound imaging and sensing system. Statistical analysis was performed to examine the test–retest reliability of the collected data, and both the main and interaction effects of each “side” (paretic vs. non-paretic) and “gait” factors, in stroke survivors. This study observed significantly good test–retest reliability in the collected data (0.794 ≤ ICC ≤ 0.985), and significant differences existed in both the side and gait factors of the average TA muscle thickness from ultrasound images, and in the gait factors of TA and MG muscle’s MMG and EMG signals (*p* < 0.05). The muscle morphological characteristics also appeared to be different between the paretic and non-paretic sides on ultrasound images. This study uncovered significantly different internal muscle contraction patterns between paretic and non-paretic sides during walking for TA (7.2% ± 1.6%) and MG (5.3% ± 4.9%) muscles in stroke survivors.

## 1. Introduction

Stroke is a leading cause of long-term disability, with an increasing prevalence and incidence due to the aging of the population globally [1]. Abnormal muscle tone and muscle weakness can cause gait asymmetry, leading to low mobility and a higher risk of falling among stroke survivors [2]. Surface electromyography (EMG) and mechanomyography (MMG) sensors can evaluate the muscle activity externally; however, the muscle’s internal contraction patterns during walking remained unknown due to the previous technical limitations. Meanwhile, the application of ultrasound imaging technologies and systems to study internal muscle activity has gained increasing interest and efforts in recent years [3,4,5,6,7,8], especially for some innovative designs of wearable ultrasound transducers and probes that have enabled the capture of ultrasound imaging in dynamic situations [9].

To comprehensively investigate the contraction patterns and activities of the tibialis anterior (TA) and medial gastrocnemius (MG) muscles of both the paretic and non-paretic legs during walking in community-dwelling stroke survivors, this study applied state-of-the-art wearable ultrasound imaging and sensing technology [9], and reported the muscle activity differences in captured ultrasound images, and EMG and MMG data. The findings of this study could provide knowledge and evidence for developing future interventions to recover lower-limb muscle function and improve the gait of stroke survivors.

## 2. Materials and Methods

### 2.1. Participants

A total of ten community-dwelling stroke survivors who could walk at least ten steps without assistive devices were recruited by the convenience sampling approach. Participants with severe visual–spatial deficits, lower-limb inflammation or fracture, or allergy to adhesive tapes were excluded from this study. Ethical approval was granted from the Institutional Review Board (IRB) of The Hong Kong Polytechnic University (Reference number: HSEARS20210623004). Written informed consent was obtained and signed by each participant. 

### 2.2. Wearable Ultrasound Imaging and Sensing System

As shown in Figure 1, the wearable ultrasound imaging and sensing system consisted of a wearable ultrasound probe (with a bandwidth of: 7.5 MHz ± 35%, and a frame rate of 10 Hz), two sets of surface EMG electrodes (272-Bx, Noraxon USA Inc., Scottsdale, AZ, USA), an MMG sensor (N1000060, VTI Technologies Oy, Vantaa, Finland), and three thin-film force sensors (A301, Tekscan Co., Ltd., South Boston, MA, USA), to simultaneously measure the muscle’s ultrasound image, electrical and mechanical activity, and the plantar force [9]. 

### 2.3. Procedure

The participant’s balance and walking ability were firstly assessed using the Berg Balance Scale (BBS) following the standard procedures [10]. Prior to each walking trial, each participant’s baseline muscle ultrasound image, for each of the four ankle muscles, was captured when the participant’s ankle joint was in a non-weight-bearing and neutral position (i.e., sitting). Participants then walked for eight meters at comfortable speed, while wearing the wearable ultrasound imaging and sensing system. When assessing the participant’s paretic TA muscle activity, the ultrasound probe was longitudinally put on the muscle belly, one set of EMG electrodes and the MMG sensor were placed in parallel beside the ultrasound probe, the other set of EMG electrodes were placed on the paretic MG muscle, and the three thin-film force sensors were placed at the first and fifth metatarsal heads and the heel on the paretic side. Similar procedures were performed when assessing the participant’s paretic MG muscle, non-paretic TA muscle, and non-paretic MG muscle. 

### 2.4. Data and Statistical Analysis

Three gait cycles in the middle of each walking trial were extracted for analysis using MATLAB (Version 2016b, The MathWorks Inc., Natick, MA, USA) [9]. The MMG and EMG data were filtered using a 4th-order Butterworth band-pass filter (5–50 Hz and 30–500 Hz, respectively), and then rectified and filtered using a moving-average filter (temporal window: 0.101 s), and finally normalized to the peak values of the three extracted gait cycles. 

For the muscle ultrasound imaging data, the upper and lower muscle boundaries were marked on each extracted ultrasound imaging frame manually by a trained practitioner after the experiment. The muscle area was then computed between the marked upper and lower muscle boundaries on each of the ultrasound imaging frames via a customized MATLAB algorithm. The muscle area was further divided by the width of the ultrasound image (i.e., 30 mm) to calculate the average muscle thickness. The three average muscle thickness values computed from the three consecutive baseline ultrasound images (where the participant’s ankle joint was in a non-weight-bearing and neutral position) were further averaged to obtain the baseline muscle thickness (determined as “100%” and used for the following normalization). Finally, the average muscle thickness on each ultrasound imaging frame was normalized to the baseline muscle thickness, by dividing the each average muscle thickness by the baseline muscle thickness, i.e., (average muscle thickness of each ultrasound imaging frame/baseline muscle thickness) × 100%. The normalized average muscle thickness was used for further data and statistical analysis in this study. 

Statistical analysis was performed using SPSS 25.0. All data were resampled to 0–100% of gait cycle with an interval of 5% [9], and the gait events/phases were identified in a gait cycle [11]. An intraclass correlation coefficient (ICC) test was performed to evaluate the test–retest reliability of the captured data. Two-way repeated ANOVA with post hoc pair-wise comparison was used to examine the main and interaction effects of “side” (paretic vs. non-paretic) and “gait” (every 5% interval in a gait cycle) factors in the captured data of participants. The level of significance was set at 0.05. 

## 3. Results

As shown in Table 1, a total of nine stroke participants completed this study (4 M + 5 F, aged 57.0 ± 8.4 years, height 160.0 ± 7.8 cm, weight 61.0 ± 10.4 kg). One participant’s data were excluded due to poor-quality images. 

### 3.1. Test–Retest Reliability Result

This study observed significantly good test–retest reliability in the captured data (*p* < 0.05): average muscle thickness (ultrasound images) for paretic TA (ICC = 0.930), non-paretic TA (ICC = 0.893), paretic MG (ICC = 0.985), non-paretic MG (ICC = 0.924); muscle mechanical activity (MMG signals) for paretic TA (ICC = 0.854), non-paretic TA (ICC = 0.834), paretic MG (ICC = 0.865), non-paretic MG (ICC = 0.865); muscle electrical activity (EMG signals) for paretic TA (ICC = 0.857), non-paretic TA (ICC = 0.772), paretic MG (ICC = 0.910) and non-paretic MG (ICC = 0.794) muscles. The overall ICC of the measurement on the four different muscles was 0.874 ± 0.0594 (mean ± standard deviation).

### 3.2. Two-Way Repeated ANOVA Result

Figure 2 shows the muscle activity changes of nine participants (mean ± standard deviation) in a gait cycle. In addition to significant interaction effects in TA muscle’s MMG data (*p* = 0.046), and in MG muscle’s EMG (*p* = 0.008) and MMG (*p* = 0.001) data, this study observed significant main effects in both the side (*p* = 0.001) and gait (*p* = 0.017) factors of the average TA muscle thickness, and in the gait factor (*p* < 0.003) of TA muscle’s EMG signal. For the captured muscle thickness and EMG/MMG data, the mean percentage differences between the paretic and non-paretic sides were 7.2 ± 1.6% and 5.3 ± 4.9% for TA and MG muscles, respectively. No significant difference existed in the gait factor of the average MG muscle thickness or TA muscle’s EMG signal.

As shown in Figure 2A,B, the average TA muscle thickness was consistently smaller at the paretic side (<100%) and larger at the non-paretic side (>100%) than the baseline average thickness (with the ankle joint in a non-weight-bearing and neutral position) throughout the gait cycle; while that of the MG muscle was consistently larger than the baseline average muscle thickness for both the paretic and non-paretic sides (>100%). The average paretic TA muscle thickness was also 5.5% smaller than that of the non-paretic side (*p* < 0.001). After heel strike, the average non-paretic MG muscle thickness decreased by 2.4% during loading response, and reached a trough (*p* = 0.034) before significantly increasing during mid-stance, and then significantly decreased by 6.4% during the initial swing; while that of the paretic side remained flat and decreased by 1.6% during terminal swing. 

Figure 3 shows the paretic and non-paretic TA and MG muscles’ ultrasound morphological characteristics in a full gait cycle for one of the stroke participants. Moving from left to right along each row, each ultrasound image illustrates the muscle’s structure and morphology during eight different stages of the gait cycle (i.e., heel strike, loading response, mid-stance, terminal stance, pre-swing, initial swing, mid-swing, and terminal swing phases, respectively) [11]. It appears that the paretic TA muscle fibers were not as organized when compared with the fibers of the other muscles (Figure 3). 

As shown in Figure 2C,D, the MMG signal of the paretic TA muscle increased significantly to 62.6% after heel strike, and reached the first peak during loading response (*p* = 0.045) and the second peak during mid-swing (32.5%, *p* = 0.016); while that of the non-paretic side increased significantly during heel strike and reached a peak during loading response (51.7%, *p* = 0.004). The paretic MG muscle’s MMG signal decreased significantly during loading response, mid-stance and mid-swing (*p* < 0.05); while that of the non-paretic side significantly increased after heel strike and reached a peak during loading response (66.7%, *p* = 0.018), followed by a significant decrease, until it increased again during pre-swing and initial swing.

As shown in Figure 2E,F, the EMG signal of the paretic TA muscle increased significantly after heel strike, and reached the first peak during loading response (49.3%, *p* = 0.009) and the second peak during pre-swing (49.9%, *p* = 0.011); while that of the non-paretic side increased significantly after heel strike, and kept fluctuating until significantly decreasing during mid-swing. The paretic MG muscle’s EMG signal increased after heel strike and reached a peak during loading response (45.1%, *p* = 0.032); while that of the non-paretic side increased significantly after heel strike, and reached the first peak during loading response (41.0%, *p* = 0.037) and the second peak during pre-swing (51.9%, *p* = 0.034).

## 4. Discussion

This study observed some interesting findings regarding the paretic and non-paretic TA and MG muscles’ activities and contraction patterns during walking in stroke survivors, using a novel wearable ultrasound imaging and sensing system [9].

The average non-paretic TA muscle thickness was 5.8% larger than that of the paretic side in a gait cycle in stroke survivors. This might be due to the muscle weakness and lack of contraction at the paretic side after stroke in patients. It is also interesting to observe that only the paretic TA muscle’s average muscle thickness was consistently smaller than (<100%) that of the baseline average muscle thickness (with the ankle joint in a non-weight-bearing and neutral position) throughout a gait cycle; while those of the non-paretic TA muscle, paretic and non-paretic MG muscles were consistently larger than (>100%) the baseline average muscle thickness. The paretic TA muscle fibers were also not as obvious or organized as those of the other three muscles, qualitatively, which may further support the observed smaller muscle thickness during walking in stroke survivors. Future studies shall verify this phenomenon and identify a quantitative parameter to describe it. This may provide more evidence for evaluating the lower-limb muscle function of patients after a stroke, and for making clinical decisions in future practice. The current wearable ultrasound imaging and sensing system can also be optimized with better image quality to capture the dynamic change in muscle architecture (e.g., fascicle length and pennation angle) quantitatively. This will provide more data and details on a stroke survivor’s leg muscle contraction patterns during walking. It is expected that with the optimized system and validated study protocol, such wearable ultrasound imaging and sensing systems may also be applied in other individuals and patients, including healthy older people, patients with osteoarthritis, cerebral palsy, amputees, etc. This will further build on our knowledge of how the lower-limb muscles contract during walking and other dynamic activities, and whether there are any differences that existed between different patient groups or not.

While the four ankle muscles’ MMG patterns and the non-paretic MG muscle’s EMG patterns were generally comparable to those of healthy adults, the paretic TA muscle’s and both the paretic and non-paretic MG muscles’ EMG patterns of stroke survivors appeared to be different [9]. However, it should be noted that the standard deviations of the EMG and MMG signals during walking were high in stroke participants. Previous studies also reported that such fluctuating patterns cannot illustrate a clear regulation of EMG changes during walking in stroke survivors [11]. Further studies involving both a stroke patient group and a healthy age-matched participant group are needed. 

## 5. Conclusions

This study applied novel wearable ultrasound imaging and sensing technology, and uncovered significantly different contraction patterns and morphological characteristics between the paretic and non-paretic sides of ankle dorsiflexors/plantar-flexors during walking in community-dwelling stroke survivors. Among the four muscles, only the paretic TA’s average muscle thickness was consistently smaller than that of the non-paretic side and the non-weight-bearing neutral position throughout a gait cycle. This builds on our knowledge of how stroke survivors’ ankle muscles contract and change internally during different events and phases of a gait cycle, and inspires further studies and evidence-based clinical practice in the field. 

## Figures and Tables

**Figure 1 biosensors-12-00349-f001:**
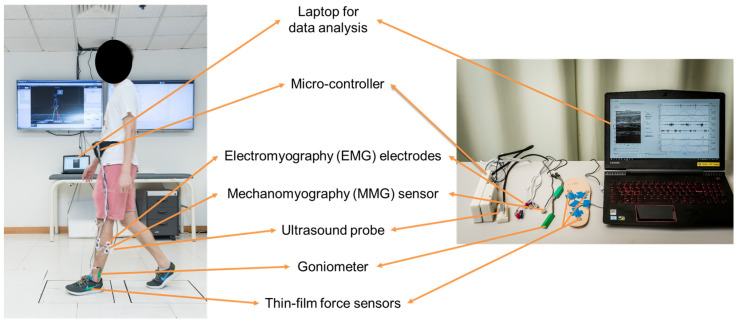
The wearable ultrasound imaging and sensing system.

**Figure 2 biosensors-12-00349-f002:**
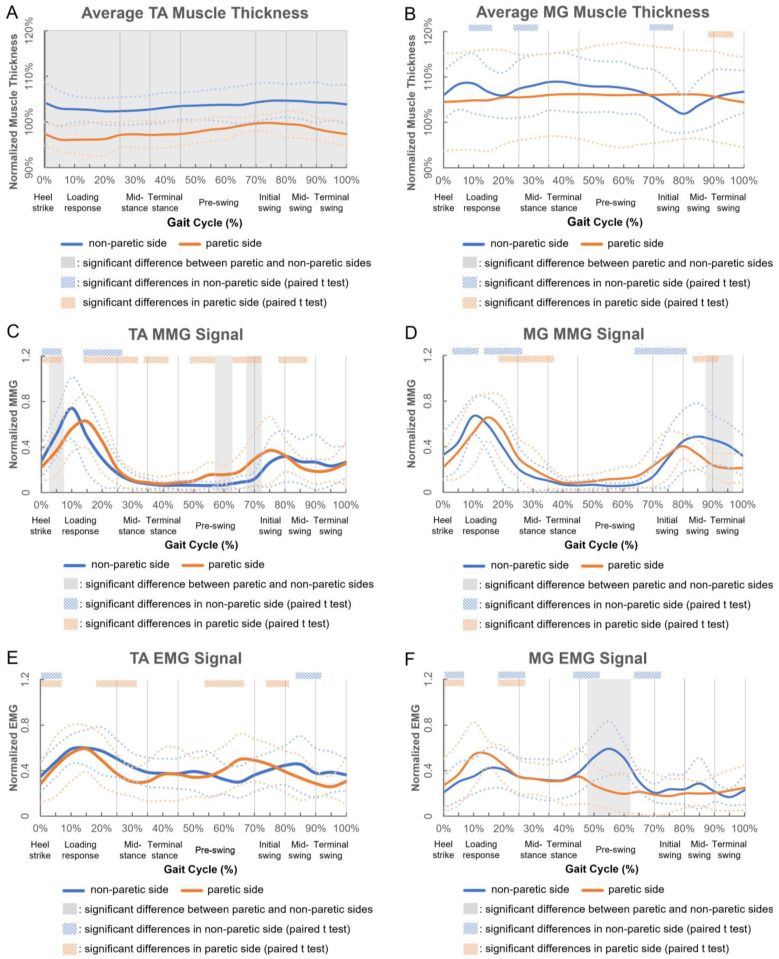
Changes in (**A**) average TA muscle thickness, (**B**) average MG muscle thickness, (**C**) mechanomyography (MMG) signal of TA muscle, (**D**) MMG signal of MG muscle, (**E**) electromyography (EMG) signal of TA muscle and (**F**) electromyography (EMG) signal of MG muscle of nine participants in a gait cycle (solid and dashed blue/orange lines indicate the mean and the standard deviation of nine participants, respectively; solid vertical gray lines indicate the typical gait events/phases of stroke survivors [11]).

**Figure 3 biosensors-12-00349-f003:**
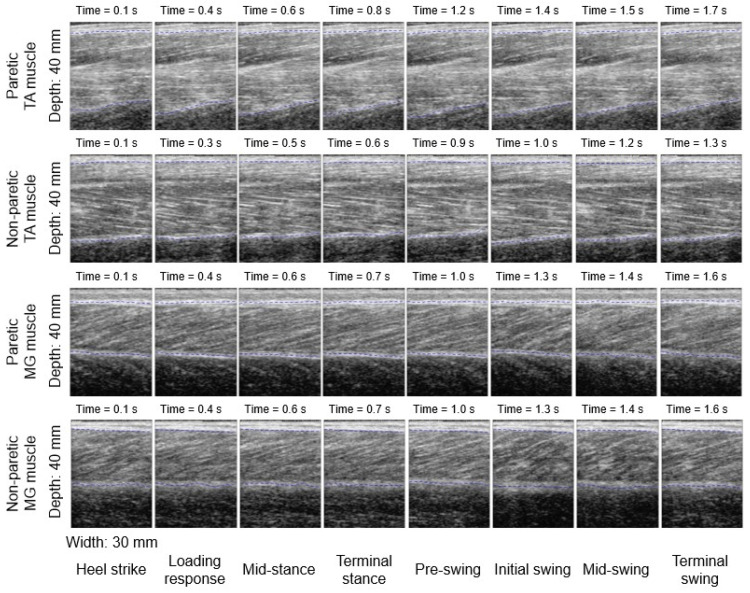
Changes in paretic and non-paretic TA and MG muscle morphological characteristics in different gait events/phases during walking in one participant.

**Table 1 biosensors-12-00349-t001:** Participants’ characteristics (n = 9).

No.	Gender	Cause of Stroke	Age	Weight (kg)	Height (cm)	Paretic Leg	BBS Score (0–56)
1	F	Hemorrhagic	60	75.2	154.0	Left	31
2	M	Ischemic	68	66.9	168.0	Right	48
3	M	Ischemic	53	78.7	172.4	Left	49
4	F	Hemorrhagic	55	55.5	156.5	Right	51
5	F	Hemorrhagic	63	51.0	150.3	Right	52
6	F	Hemorrhagic	39	49.1	155.3	Right	53
7	M	Hemorrhagic	55	63.0	166.0	Right	55
8	M	Ischemic	61	54.8	152.0	Left	56
9	F	Ischemic	63	58.2	163.0	Right	56

## Data Availability

The data presented in this study are available within this article.

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
