# Peer review of "How Paretic and Non-Paretic Ankle Muscles Contract during Walking in Stroke Survivors: New Insight Using Novel Wearable Ultrasound Imaging and Sensing Technology"

_biosensors, 2022, doi:10.3390/bios12050349_

Round 1

Reviewer 1 Report

This report investigate the paretic and non-paretic TA and MG muscles’ MMG, EMG, and muscle ultrasound image of stroke survivors, and discussed the difference of muscle internal contraction patterns. The wearable ultrasound imaging and sensing technology is novel and could be applicated in more physical conditions and body parts, especially be used in medical and rehabilitationto field.

Some suggestions:

  1. In figure 1, the location of the interval ruler should be marked on the axis, especially the location of normalized 100%, which will be helpful to get comparative results.
  2. On line 76-77, “The upper-/lower- muscle boundaries were marked on each extracted frame to compute muscle area, then divided by the width of image to calculate the averaged muscle thickness”, which kind of frame is the extracted frame mentiond here, is it the ultrasound frame? What’s the value of thickness of baseline muscle?
  3. I noticed that the thickness of muscle in each moment of figure 2 is different from left to right. How could you got the value of thickness from each picture?

Author Response

Point-to-point response to editor's and reviewers’ comments

We would like to thank the editor and reviewers for their constructive feedback and believe the manuscript is stronger as a result. Below we detail the changes to the manuscript made to address the editor’s and reviewers’ comments/concerns.

Reviewer 1

Thank you for opportunity to review the brief report entitled “How paretic and non-paretic ankle muscles contract during walking in stroke survivors? New insight from novel wearable ultrasound imaging and sensing technology”.

The aim of the study was to uncover the muscle activity and contraction pattern of tibialis anterior (TA) and medial gastrocnemius (MG) of both the paretic and non-paretic sides during walking in stroke survivors by using a state-of-the-art wearable ultrasound imaging and sensing system. The study was performed on 10 stroke survivors. Based on the results from the conducted study, the paretic TA muscle thickness was consistently smaller than that of non-paretic side and non-weight-bearing neutral position in a gait cycle. While the ankle muscles’ MMG patterns and non-paretic MG muscle’s EMG patterns were generally comparable/similar to those of healthy adults, the paretic TA and paretic/non-paretic MG muscles’ EMG patterns of stroke survivors appeared to be different.

This report investigates the paretic and non-paretic TA and MG muscles’ MMG, EMG, and muscle ultrasound image of stroke survivors, and discussed the difference of muscle internal contraction patterns. The wearable ultrasound imaging and sensing technology is novel and could be applicated in more physical conditions and body parts, especially be used in medical and rehabilitation field.

  1. In figure 1, the location of the interval ruler should be marked on the axis, especially the location of normalized 100%, which will be helpful to get comparative results.

As suggested, the location of the interval ruler was added in the figures in line 175 as below:

Figure 2. Percentage changes in (a) averaged TA muscle thickness, (b) averaged MG muscle thickness, (c) mechanomyography (MMG) signal of TA muscle, (d) MMG signal of MG muscle, (e) electromyography (EMG) signal of TA muscle and (f) electromyography (EMG) signal of MG muscle of nine participants in a gait cycle. (Dashed blue/orange lines indicate the standard deviation (SD) among participants; Solid vertical gray lines indicate the typical gait events/phases of stroke survivors [11].)

  1. On line 76-77, “The upper-/lower- muscle boundaries were marked on each extracted frame to compute muscle area, then divided by the width of image to calculate the averaged muscle thickness”, which kind of frame is the extracted frame mentioned here, is it the ultrasound frame?

Thanks for asking. “The extracted frame” was referred to the ultrasound-imaging frame.

To clarify this, the description has been revised in line 91-92, stating: “…were marked on each extracted ultrasound-imaging …”

What’s the value of thickness of baseline muscle?

The baseline value of muscle thickness was the average thickness calculated from three consecutive ultrasound frames which were captured when participant’s ankle joint was in non-weight-bearing and neutral position. And this baseline value was set as 100%.

Such clarification has been added:

  • in line 72-74, stating: “…Prior to each walking trial, each participant’s baseline muscle ultrasound image, for each of the four ankle muscles, was captured when the participant’s ankle joint was in non-weight-bearing and neutral position.…”

  • in line 97-106, stating: “…The three averaged muscle thickness values computed from the three consecutive baseline ultrasound images (where the participant’s ankle joint was in non-weight-bearing and neutral position), were further averaged to get the baseline averaged muscle thickness value (determined as “100%” and used for the following normalization). Finally, the computed averaged muscle thickness value on each ultrasound-imaging frame was normalized to the baseline averaged muscle thickness, by dividing the corresponding computed averaged muscle thickness value by the baseline averaged muscle thickness [i.e., (computed averaged muscle thickness value of each ultrasound-imaging frame / baseline averaged muscle thickness) * 100%]. The normalized averaged muscle thickness values were used for further data and statistical analysis in this study…”
  1. I noticed that the thickness of muscle in each moment of figure 2 is different from left to right. How could you get the value of thickness from each picture?

Thanks for asking. For each row of the ultrasound images, each frame of ultrasound image is corresponding to the muscle structure/morphology of different gait events in a gait cycle. From left to right, each ultrasound image illustrates the muscle structure/morphology in gait events of heel strike, loading response, mid-stance, terminal stance, pre-swing, initial swing, mid-swing, and terminal swing phase.

To better clarify this and avoid potential misunderstanding in readers, more details were added in line 148-153, stating: “… An example of the paretic and non-paretic TA and MG muscles’ ultrasound morphological characteristics in a full gait cycle of one stroke participant is illustrated in Figure 3. For each row of the muscle ultrasound images and from left to right, each ultrasound image illustrates the corresponding muscle ultrasound-imaging structure/morphology in gait events of heel strike, loading response, mid-stance, terminal stance, pre-swing, initial swing, mid-swing, and terminal-swing phases, respectively [11]…”

Reviewer 2 Report

The article deals with the application of wearable ultrasound imaging and sensing technology on stroke survivors. The article is very interesting to me. Generally, the paper has a clear aim and motivation, well-explained assessment methods, results, and discussion.  The results presented in the article are original and interesting. The conclusions are consistent with the results provided in the graphs.

I have the following minor comments on the paper:

I would appreciate a photo of a wearable measuring system.

Line 75: “The upper-/lower- muscle boundaries were marked on each extracted frame to compute muscle area”. Please specify if the boundaries were marked automatically or manually.

Author Response

Point-to-point response to editor's and reviewers’ comments

We would like to thank the editor and reviewers for their constructive feedback and believe the manuscript is stronger as a result. Below we detail the changes to the manuscript made to address the editor’s and reviewers’ comments/concerns.

Reviewer 2

The article deals with the application of wearable ultrasound imaging and sensing technology on stroke survivors. The article is very interesting to me. Generally, the paper has a clear aim and motivation, well-explained assessment methods, results, and discussion.  The results presented in the article are original and interesting. The conclusions are consistent with the results provided in the graphs.

  1. I would appreciate a photo of a wearable measuring system.

As suggested, a new figure illustrating the wearable measuring system was added in line 68-69 as below:

Figure 1. The Wearable Ultrasound Imaging and Sensing System

  1. Line 75: “The upper-/lower- muscle boundaries were marked on each extracted frame to compute muscle area”. Please specify if the boundaries were marked automatically or manually.

Thanks for pointing this out. The upper-/lower- muscle boundaries of each ultrasound frame were marked manually by a trained practitioner.

Such information was added in the manuscript in line 91-93, stating: “For the muscle ultrasound imaging data, the upper-/lower- muscle boundaries were marked on each extracted ultrasound-imaging frame manually by a trained practitioner firstly after the experiment.”
